# Who gets referred for knee or hip replacement? A theoretical model of the potential impact of evidence-based referral thresholds using data from a retrospective review of clinic records from an English musculoskeletal referral hub

Helen A Dakin [1], Peter Eibich [1,2], Alastair Gray [1], James Smith,[3,4] Karen L Barker,[3] David Beard,[3] Andrew J Price [3] on behalf of the ACHE Study team

For numbered affiliations see end of article.

**Correspondence to**
Dr Helen A Dakin;
helen.dakin@ndph.ox.ac.uk

## ABSTRACT

**Objectives** To estimate the relationship between patient characteristics and referral decisions made by musculoskeletal hubs, and to assess the possible impact of an evidence-based referral tool.

**Design** Retrospective analysis of medical records and decision tree model evaluating policy changes using local and national data.

**Setting** One musculoskeletal interface clinic (hub) in England.

**Participants** 922 adults aged ≥50 years referred by general practitioners with symptoms of knee or hip osteoarthritis.

**Interventions** We assessed the current frequency and determinants of referrals from one hub and the change in referrals that would occur at this centre and nationally if evidence-based thresholds for referral (Oxford Knee and Hip Scores, OKS/OHS) were introduced.

**Main outcome measure** OKS/OHS, referrals for surgical assessment, referrals for arthroplasty, costs and quality-adjusted life years.

**Results** Of 110 patients with knee symptoms attending face-to-face hub consultations, 49 (45%) were referred for surgical assessment; the mean OKS for these 49 patients was 18 (range: 1–41). Of 101 hip patients, 36 (36%) were referred for surgical assessment (mean OHS: 21, range: 5–44). No patients referred for surgical assessment were above previously reported economic thresholds for OKS (43) or OHS (45). Setting thresholds of OKS ≤31 and OHS ≤35 might have resulted in an additional 22 knee referrals and 26 hip referrals in our cohort. Extrapolating hub results across England suggests a possible increase in referrals nationally, of around 13 000 additional knee replacements and 4500 additional hip replacements each year.

**Conclusions** Musculoskeletal hubs currently consider OKS/OHS and other factors when making decisions about referral to secondary care for joint replacement. Those referred typically have low OHS/OKS, and introducing

## Strengths and limitations of this study

► This is the first analysis reporting data on the characteristics of patients being referred to a UK hub with symptoms of hip or knee osteoarthritis; such data are needed to assess the impact of changing referral criteria on the number of knee/hip replacements and on costs and quality of life.

► We retrospectively reviewed clinic records for 922 men and women referred to a single musculoskeletal hub, although only 221 patients underwent arthroplasty.

► We used several assumptions to extrapolate our estimates of how the probability of referral varies with preoperative characteristics using a decision tree and Markov model, which enabled us to estimate the potential impact of different referral policies locally and nationally.

► A prospective pilot study would be required to quantify the real-world impact of any policy change, while data from multiple hubs and information on other patient characteristics (such as body mass index) are required for a comprehensive understanding of current clinical practice.

evidence-based OKS/OHS thresholds would prevent few inappropriate (high-functioning, low-pain) referrals. However, our findings suggest that some patients not currently referred could benefit from arthroplasty based on OKS/OHS. More research is required to explore other important patient characteristics currently influencing hub decisions.

## INTRODUCTION

Many clinical commissioning groups (CCGs) in the UK have set referral criteria for joint



replacement that exclude patients with Oxford Knee and Hip Scores (OKS and OHS) above certain thresholds, which are often as low as 19[1] or 24[1–3] out of 48. Recent papers have shown that these low thresholds are inappropriate: arthroplasty has a ≥80% chance of producing a meaningful improvement in patients with OKS/OHS of 19 or 24[4] and would cost <£10 000 per quality-adjusted life year (QALY).[5] The recent Arthroplasty Candidacy Help Engine (ACHE) study analysed >400 000 medical records and concluded that referral thresholds between 31 and 41 may be justified on clinical and cost-effectiveness grounds.[4 5]

Commissioners' decisions concerning thresholds may be influenced by the likely number of operations and their cost. In the UK, >94% of patients currently undergoing arthroplasty have OKS/OHS ≤31, demonstrating that nearly all operations currently conducted are cost-effective and have ≥70% chance of meaningful improvement.[4 5] However, there are very little data on how patient characteristics influence the assessment process in practice, or on patients who do not currently receive surgery, making it difficult to assess the impact that different referral criteria may have on patient numbers or costs. As in many other clinical areas, hubs (also known as triage clinics, musculoskeletal interface services or intermediate musculoskeletal assessment centres) are increasingly used as gatekeepers determining access to consultations with orthopaedic surgeons, assessing all patients being considered for surgery.[6–8]

We aimed to review the characteristics of patients attending hub consultations, estimate the proportion of patients referred for surgical assessment and surgery, and explore how referral decisions vary with preoperative characteristics. To illustrate how such data could be used to inform policy, we estimated the potential impact of a change in referral criteria, namely basing referrals from the hub to surgical assessment on evidence-based OKS/OHS thresholds, rather than the hub's current referral criteria.

## METHODS
### Outline
We collected data from one UK referral hub and used them to estimate a logistic regression model predicting the probability that patients will be referred from the hub to surgical assessment based on their OKS/OHS, age and sex. We then subsequently applied this regression model to nationally collected preoperative outcomes data[9–14] to estimate how many patients are referred for surgical assessment nationally. We predicted how many of these patients would be referred through to secondary care if different OKS/OHS thresholds were introduced at the hub and potentially how many extra operations would be performed.

### Hub data collection
Anonymised data were extracted retrospectively from medical records of patients with knee or hip symptoms who had been referred by general practitioners (GPs) between July 2015 and July 2016 to the musculoskeletal hub at the Nuffield Orthopaedic Centre (NOC) in Oxford. This study was discussed and agreed through the Clinical Governance Group for hip and knee replacement as part of a wider review of outcomes in hip and knee arthroplasty. The primary statistical analysis used a prognostic model to estimate how the probability of patients attending face-to-face consultations at the hub being referred to surgical assessments in secondary care varied with OKS/OHS, age and sex. We therefore chose the sample size to provide data on ≥30 knee referrals and ≥30 hip referrals from the hub to secondary care, thereby providing ≥10 events per explanatory variable.[15]

Two medically qualified surgical research fellows extracted the following: age, sex, OKS/OHS at hub attendance, attendance date, whether the patient was referred to secondary care, date of any subsequent surgical assessment visit and date of any subsequent arthroplasty surgery. Additional information on imaging, referrals to other clinics, and other surgeries or diagnoses was recorded in free-text fields. Data on OKS at the surgical assessment visit were also extracted, when available, from the secondary care records of patients with knee pain. Body mass index (BMI) was not extracted as the analysis focused on OKS/OHS and the available evidence on how capacity to benefit and cost-effectiveness vary with OKS/OHS does not consider BMI.[4 5] Patients who were on the waiting list for arthroplasty surgery at the time of data extraction (August 2016) and those for whom surgery was delayed due to comorbidities or high BMI after their attendance at the surgical assessment consultation were counted as having been referred for surgery.

Exclusion criteria comprised the following:
1. Aged <50 years (for whom knee/hip pain is unlikely to be caused by osteoarthritis).
2. Evidence from medical records that symptoms were due to a condition other than osteoarthritis.
3. Previous arthroplasty on the same joint.
4. Medical records inaccessible for research.
5. Any attendance at the hub or surgical assessment unit before July 2015.

However, patients who were referred for X-rays, MRI or physiotherapy but did not attend face-to-face consultations at the hub or surgical assessment unit were included in the descriptive analysis.

### Statistical modelling
We used logistic regression to estimate a prognostic model of the local hub data that predicted how OKS/OHS, age and sex affected the odds of being referred for surgical assessment following a face-to-face attendance at the hub. Age and OKS/OHS at hub attendance were analysed as continuous variables. Explanatory variables were selected based on the Akaike information criterion using forward stepwise regression, manually testing variables in a prespecified sequence (online supplementary appendix). Analyses were conducted in Stata V.14 on a complete case

basis, excluding patients with missing data. We analysed binary variables using two-sample tests of proportions and analysed continuous variables using unpaired t-tests preceded by *F*-tests for equal variance. OKS/OHS in the hub sample was compared against population means[9–14] using one-sample t-tests.

## Estimating the number of referrals and effect of introducing thresholds

A decision tree model of the treatment pathway was developed based on hub data and the authors' clinical experience of running hubs and/or surgical clinics. The overall proportion of patients referred directly to surgical assessment and the proportion attending face-to-face hub consultations were calculated from the hub sample. We also estimated the proportion of patients who underwent arthroplasty after being referred from the hub to surgical assessment.

The model then extrapolated hub data to estimate the number of patients in England who are referred for hip or knee replacement (online supplementary appendix). Logistic regression results were used to predict the probability of being referred from the hub to surgical assessment for men and women aged 50, 60, 70, 80 and 90 with different OKS/OHS. For each patient group defined by age, sex and OKS/OHS, we then estimated the number of patients referred to hubs nationally by dividing national data on the number of joint replacements currently conducted for osteoarthritis in England[9–14] by the probability that patients in each group would undergo surgery.

We then conducted an exploratory analysis estimating how the number of referrals to surgical assessment and the number of arthroplasty procedures might change if different OKS/OHS thresholds between 18 and 45 were used to determine referral decisions during face-to-face hub consultations. Although in practice thresholds could be used at various stages in the referral pathway, our analysis focused on the impact of changing referral criteria during face-to-face hub consultations since OKS/OHS data were only routinely available for this patient group.

This analysis made the following assumptions:

1. Introducing OKS/OHS thresholds at the hub was assumed to have no effect on GP referrals, the hub triage or the probability of a patient with particular preoperative characteristics being referred for arthroplasty following a surgical assessment visit.
2. Since patient records provided no data on OKS/OHS for patients who did not attend the hub, we assumed that the probability of direct referral to surgical assessment and the probability of attending a face-to-face visit at the hub were independent of OKS/OHS, age and sex. Although this assumption may not hold in practice (as symptom severity is one of the main factors considered in the hub triage), it is unlikely to affect estimates of the impact of changing referral criteria solely at the hub.
3. We also assumed that the probability of patients referred to surgical assessment subsequently being referred for surgery was independent of OKS/OHS, age and sex, which was supported by a secondary regression analysis (online supplementary appendix).
4. We assumed that the referral probabilities estimated from the NOC hub sample are broadly representative of clinical practice across the UK.
5. We assumed, based on the experience of clinical co-authors, that 50% of patients who are currently not referred for surgical assessment after a face-to-face hub visit would still not be considered candidates for arthroplasty (and therefore not referred) regardless of whether thresholds were introduced. This group of patients includes patients who choose not to be referred, those who have not had a complete trial of non-operative treatment and others who are unsuitable for surgery due to comorbidities or other factors. The remaining 50% of patients were assumed to be referred only if their OKS/OHS was below the threshold.
6. Since there are no published data on how OKS/OHS changes over time in the absence of arthroplasty, we assumed for simplicity patients who do not undergo arthroplasty following their hub attendance would not be referred back to clinic for 10 years.
7. Each 40 min hub attendance was estimated to cost £58 and surgical assessment £132 (online supplementary appendix, online supplementary table A1).
8. The analyses took the perspective of the National Health Service (NHS), focusing on costs related to knee/hip arthroplasty. Costs of GP consultations, hub triage, X-rays, imaging, physiotherapy, injections, weight loss programmes, missed appointments and referrals to other clinics were excluded as there is no reason to expect the proportion of patients requiring these services to change following the introduction of OKS/OHS thresholds.
9. The reference year for costs was 2014.

The patient numbers calculated within the decision tree were used to estimate the cost of the referral pathway. The total 10-year costs and total QALYs estimated in a related study[5] for different patient characteristics with and without arthroplasty were applied to the number of patients expected to undergo arthroplasty or have no arthroplasty in each scenario. These figures were used to estimate the net health benefit for each scenario[16 17] (assuming that the NHS is willing/able to pay £20 000 per QALY gained[18]) and the incremental cost-effectiveness of different pairs of scenarios (online supplementary appendix).

We also estimated the number of additional referrals for surgical assessment that might have been observed within the population of individuals attending the NOC hub if a fixed OKS/OHS threshold was introduced. Based on assumption 5, this was equal to half the number of patients who had OKS/OHS at or below the proposed threshold but were not referred, plus the number who were referred with OKS/OHS below the proposed threshold: for example, if (hypothetically) 60 out of 100 patients were referred and 55 of those referred and 24 of those not referred had OKS <31,

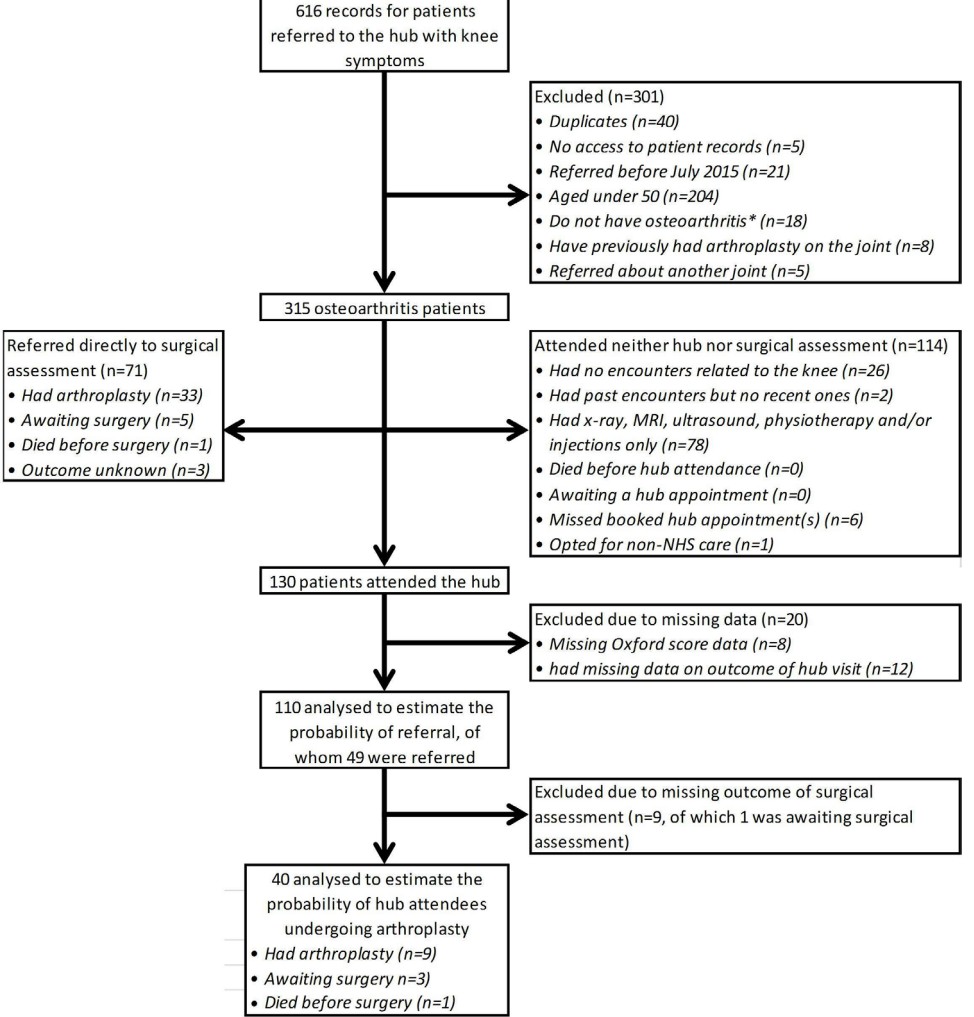

**Figure 1** Patient flow diagram for patients referred with knee symptoms. *See online supplementary table A2 for a list of the conditions other than osteoarthritis for which patients aged ≥50 years were excluded from the analysis. NHS, National Health Service.

we would estimate that 67 (55+24/2) would be referred if a threshold of 31 was introduced.

### Public and patient involvement

Patient representatives were involved in the grant application and design of the wider ACHE study.[4] The ACHE research question and study design (prior to funding) were informed by patient interest groups. The current analysis was based on pre-existing data from medical records so no patients were directly recruited in the course of the work. Throughout the ACHE study, a user group, including patients and other stakeholders, informed progress and development of the work. Results of the analysis were discussed with user group, including clinicians and patient representatives, to inform the presentation of results.

### RESULTS
#### Current referral pathway

From the 1638 records reviewed, we identified 315 patients with knee osteoarthritis and 607 with hip osteoarthritis:

922 in total (figures 1 and 2, online supplementary figure A1, online supplementary tables A2-A3). Data on these patients were used to construct the treatment pathway shown in figure 3.

During the time period for which data were collected, the musculoskeletal hub service was the only route for patients in Oxfordshire to access NHS elective orthopaedic surgery and also directed patients to physiotherapy in the absence of a direct referral physiotherapy service. When GPs refer patients to secondary care with knee/hip symptoms, the referral letter and X-rays are first reviewed by senior hub staff, who decide whether the patient should (1) be referred directly to surgical assessment, (2) attend the hub clinic or (3) be managed in primary care (figure 3). Triage is based on the symptoms described in the GP's referral letter (which will include BMI and may include OKS/OHS) and the conservative treatments already given (such as anti-inflammatory drugs, physiotherapy and weight loss strategies).

Of patients with osteoarthritis included in the analysis, 23% (71 of 315) of knee patients and 39% (236 of 607) of

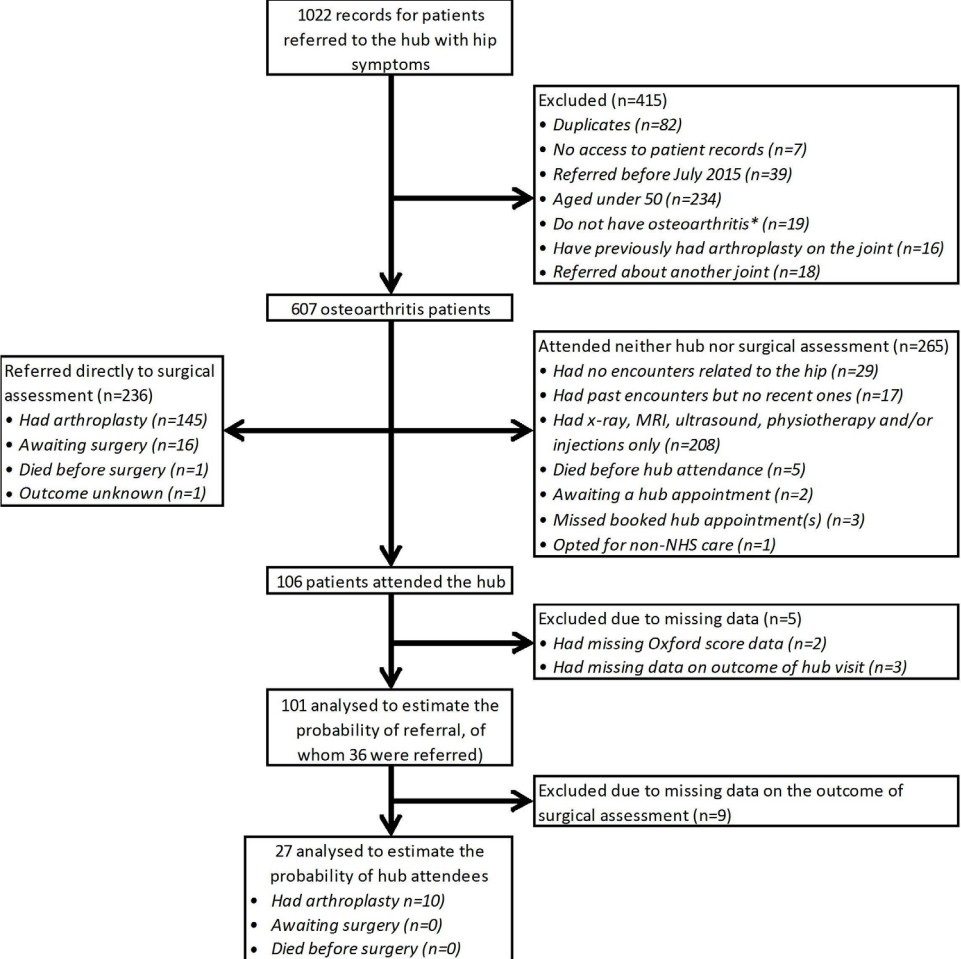

**Figure 2** Patient flow diagram for patients referred with hip symptoms. *See online supplementary table A2 in online supplementary appendix for a list of the conditions other than osteoarthritis for which patients aged ≥50 years were excluded from the analysis. NHS, National Health Service.

hip patients were referred directly to surgical assessment based on the hub triage. In the authors' experience, these patients tend to be those with severe symptoms, those who had already exhausted all conservative measures and those who had previously been referred for surgical assessments, but had chosen not to have surgery at that time. Among 27 patients referred directly to surgical assessment for whom OKS data were available, the mean OKS was 15 (range: 1–30); no such data were available for hip patients.

At the surgical assessment visit, patients discuss the risks and benefits of surgery with an orthopaedic surgeon and make an informed decision about whether or not to undergo arthroplasty or other surgery, taking into account their comorbidities, symptom severity and other factors. Those not undergoing arthroplasty may have interventional radiology or other operations. Among the patients who were referred directly for surgical assessment and had outcomes data available, 56% (38 of 68) of knee patients and 69% (161 of 235) hip patients underwent or were awaiting knee/hip replacements at the time of data extraction.

Following triage, 36% (114 of 315) of knee patients and 44% (265 of 607) of hip patients did not attend face-to-face consultations at either the hub or surgical assessment. Typically such patients comprise those who can be managed in primary care: for example those with mild symptoms and those who have not yet exhausted conservative treatment options, such as advice and information, activity and exercise and weight loss. Across the 379 hip and knee patients without face-to-face consultations, 270 (71%) had X-rays, MRI or ultrasound to identify the most appropriate care pathway. Eleven (3%) were referred for physiotherapy, while seven (2%) patients, all with hip osteoarthritis, had injections. Other patients may not be seen face-to-face as they are unfit for surgery or have recent injuries likely to heal without further intervention. Eighteen patients opted for non-NHS care, missed/failed to book hub appointments, died before the hub attendance or were still awaiting a hub appointment at the time of data extraction (figures 1 and 2).

The remaining 41% (130 of 315) of knee patients and 17% (106 of 607) of hip patients attended face-to-face assessments at the musculoskeletal hub, where

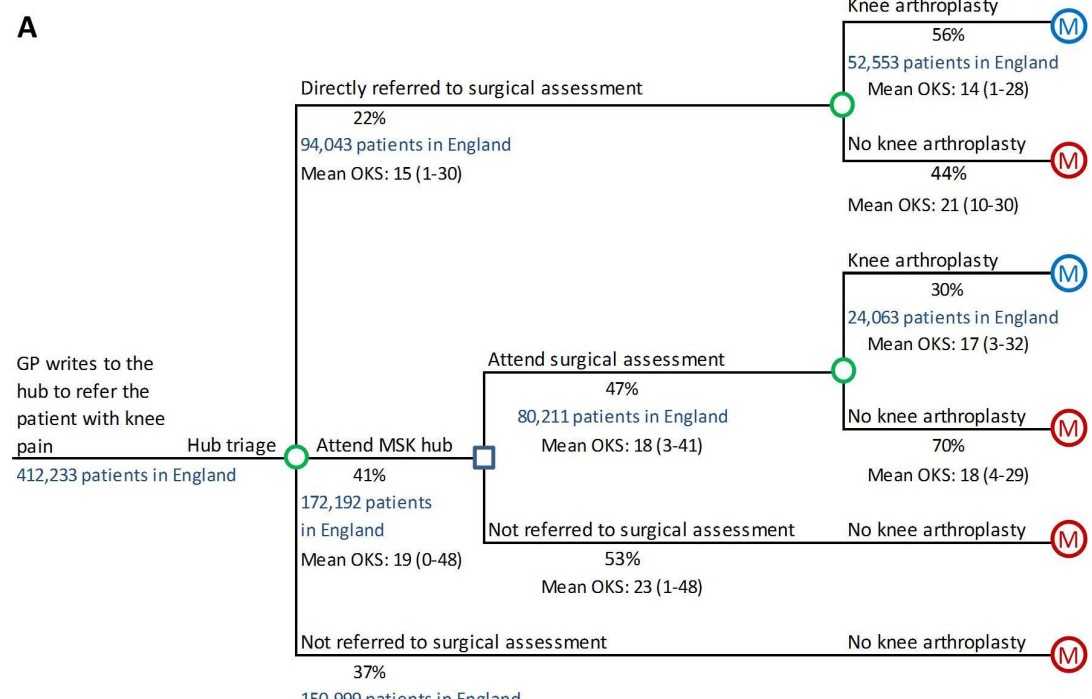

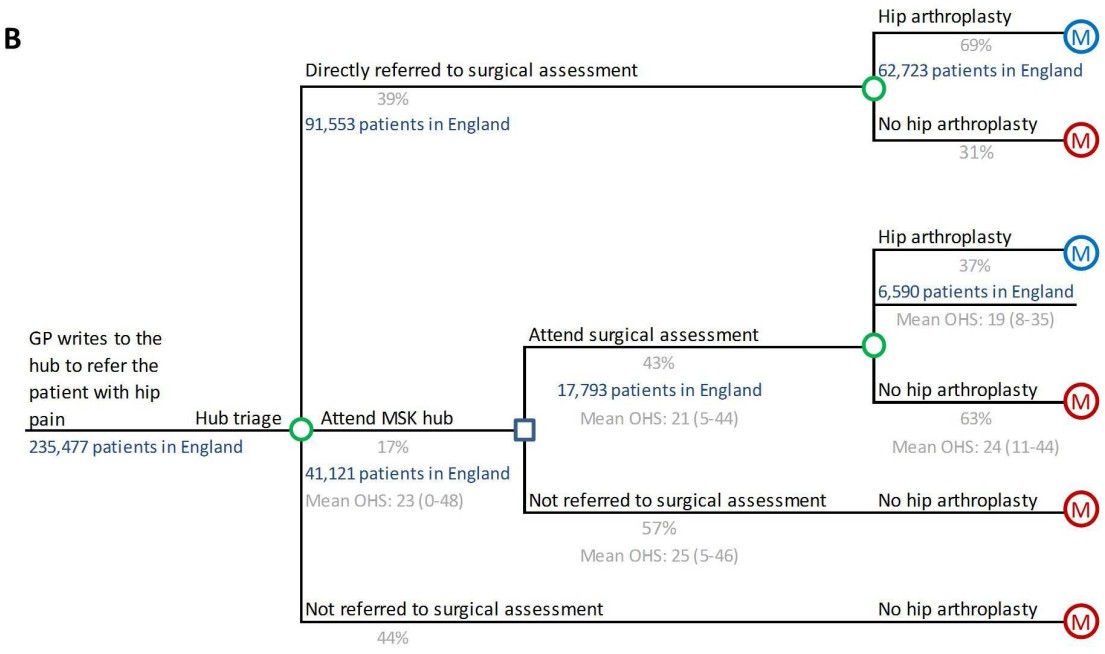

**Figure 3** Number of patients predicted to be referred with (A) knee and (B) hip osteoarthritis symptoms in England and mean Oxford Knee and Hip Score (OKS/OHS) for the groups for which data are available. GP, general practitioner; MSK, musculoskeletal.

extended-scope specialist physiotherapists or orthopaedic fellows assess patients to confirm diagnosis. Patients routinely complete OKS/OHS questionnaires to assess whether the level of symptoms warrants surgery and to guide discussions about the symptom profile. Among the patients attending the hub, the mean OKS was 21 (range: 1–48; online supplementary table A3, online supplementary figure A1), while the mean OHS was 24 (range: 5–46);

both values were significantly higher than the average for patients undergoing arthroplasty nationally (mean: 18; p≤0.015), although only OHS was significantly different from the average for patients undergoing arthroplasty in Oxfordshire (mean OKS: 20, p=0.134; mean OHS: 19, p<0.0001).[9–14]

Diagnostic imaging, landmark injections and injections for trochanteric bursitis may be done during the hub

consultation. Patients with BMI ≥40 are referred for monitored weight loss programmes that must be followed for 12 months before considering surgery. Staff and patients also discuss the risks and benefits of joint replacement (including recovery times, the need for support at home after hospital discharge and the potential need for revision surgery) and how these may be affected by patients' living arrangements and comorbidities. Following the hub visit, 45% (49 of 130) of knee patients and 36% (36 of 101) of hip patients were referred for surgical assessment. This includes patients who are considered candidates for arthroplasty or other procedures (eg, arthroscopy, anterior cruciate ligament repair or interventional radiology).

Patients with higher OKS/OHS were significantly less likely to be referred for surgical assessment: each one-point increase in OKS reduced the odds of referral by 4.7% (p=0.019), while a one-point increase in OHS reduced the odds by 3.9% (p=0.062; online supplementary table A4). Those patients who are not referred to surgical assessment may be referred for physiotherapy, other non-surgical management or other outpatient clinics (eg, rheumatology or sports injury clinics), or may choose not to be referred as they prefer not to undergo surgery at the current time.

Across the hub attendees who were referred for surgical assessment and had data on clinical outcomes, 30% (12 of 40) of knee patients and 37% (10 of 27) of hip patients underwent or were awaiting arthroplasty surgery.

Applying estimates of the probability of referral and subsequent arthroplasty from the hub data set to national data on the distribution of patients undergoing knee/hip arthroplasty suggests that GPs in England refer around 417 000 patients with knee osteoarthritis and around 235 000 patients with hip osteoarthritis to secondary care each year (figure 3). Of these, around 172 000 knee patients and 41 000 hip patients might attend a hub if the care pathway in Oxfordshire were followed nationally; such hub attendances would cost a total of £12 million.

## Effect of introducing referral thresholds
### Knees
We used the hub data to estimate the impact that using OKS to guide referral decisions during face-to-face hub consultations might have on patient numbers, costs and health outcomes. Table 1 shows the results for thresholds between 18 and 43, although we focus here on the impact of an OKS threshold of 31, since the ACHE study reported that people with preoperative OKS ≤31 have a ≥70% chance of achieving a seven-point increase in OKS following knee arthroplasty.[4]

Of the 110 knee patients attending the NOC hub, 94 (85%) had OKS ≤31, of whom 47 (50%) were referred for surgical assessment. Two patients were referred with OKS of 32 or 41. Assuming that 50% of the NOC patients with OKS ≤31 who are not currently referred might still choose not to be referred or might be considered clinically inappropriate for other reasons, a fixed referral threshold OKS of 31 would have resulted in an additional 22 referrals to surgical assessment: a 49% increase.

Applying the results of logistic regression to the whole population undergoing surgery nationally suggested that an OKS threshold of 31 might result in 43 000 additional surgical assessment visits and 13 000 additional knee replacement procedures in England each year (table 1).

**Table 1** Estimates of the potential impact of different OKS thresholds on patient numbers, costs and QALYs among the 172 192 patients with knee osteoarthritis attending the hub in England each year

| | Current practice | Maximum OKS at which patients can be referred for surgical assessment | | | | | |
| --- | --- | --- | --- | --- | --- | --- | --- |
| | | 18 | 24 | 31* | 35 | 41† | 43‡ |
| Number of attendances at the surgical outpatient visit | 80 211 | 65 286 | 99 982 | 123 635 | 129 256 | 131 810 | 131 983 |
| Number of arthroplasty procedures conducted (% change§) | 24 063 | 19 586 (−6%) | 29 995 (+8%) | 37 090 (+17%) | 38 777 (+19%) | 39 543 (+20%) | 39 595 (+20%) |
| Total cost over 10 years (thousands) | £1 098 213 | £1 081 313 | £1 108 836 | £1 134 134 | £1 142 073 | £1 146 346 | £1 146 678 |
| Total QALYs over 10 years | 564 744 | 566 658 | 585 674 | 594 802 | 596 910 | 597 825 | 597 856 |
| Net health benefit (QALYs)¶ | 509 834 | 512 593 | 530 232 | 538 096 | 539 806 | 540 508 | 540 522 |

The results presented exclude patients who did not attend face-to-face consultations at the hub; based on our analysis, 31% (24 063 of 76 617) of knee replacements are conducted on patients who attended the hub.
*Threshold at which 70% of patients are predicted to achieve a seven-point improvement in OKS.[4]
†Arthroplasty Candidacy Help Engine (ACHE) absolute threshold, above which patients cannot achieve a seven-point improvement in OKS.[4]
‡ACHE economic threshold, above which arthroplasty is not cost-effective (ie, costs >£20 000 per QALY gained).[5]
§Percentage change in the total number of arthroplasty procedures following a change to referral patterns at the hub. Equal to the difference in the number of procedures between the scenario in question and 'current practice', divided by the 76 617 knee replacements conducted in England each year.[23]
¶Net health benefit=QALYs – cost/£20 000, and indicates the QALYs for each scenario, minus the health benefits that would be foregone by spending money on knee arthroplasty candidates, rather than other conditions.
OKS, Oxford Knee Score; QALY, quality-adjusted life year.

**Table 2** Estimates of the potential impact of different OHS thresholds on patient numbers, costs and QALYs among the 41 121 patients with hip osteoarthritis attending the hub in England each year

| | Current practice | Maximum OHS at which patients can be referred for surgical assessment | | | | | |
|---|---|---|---|---|---|---|---|
| | | 18 | 24 | 30 | 35* | 40† | 45‡ |
| Number of attendances at the surgical outpatient visit | 17 793 | 18 036 | 25 420 | 29 871 | 31 561 | 32 108 | 32 216 |
| Number of arthroplasty procedures conducted (% change§) | 6590 | 6680 (+0%) | 9415 (+4%) | 11 063 (+6%) | 11 689 (+7%) | 11 892 (+8%) | 11 932 (+8%) |
| Total cost over 10 years (thousands) | £168 402 | £168 202 | £182 080 | £190 519 | £193 753 | £194 816 | £195 030 |
| Total QALYs over 10 years | 123 674 | 128 990 | 135 863 | 138 888 | 139 929 | 140 248 | 140 288 |
| Net health benefit (QALYs)¶ | 115 254 | 120 580 | 126 759 | 129 362 | 130 242 | 130 508 | 130 537 |

The results presented exclude patients who did not attend face-to-face consultations at the hub; based on our analysis, 9% (6590 of 69 313) of hip replacements are conducted on patients who attended the hub.
*Threshold at which 70% of patients are predicted to achieve an eight-point improvement in OHS.[4]
†Arthroplasty Candidacy Help Engine (ACHE) absolute threshold, above which patients cannot achieve an eight-point improvement in OHS.[4]
‡ACHE economic threshold, above which arthroplasty is not cost-effective (ie, costs >£20 000 per QALY gained).[5]
§Percentage change in the total number of arthroplasty procedures following a change to referral patterns at the hub. Equal to the difference in the number of procedures between the scenario in question and 'current practice', divided by the 69 313 hip replacements conducted in England each year.[23]
¶Net health benefit=QALYs – cost/£20 000, and indicates the QALYs for each scenario, minus the health benefits that would be foregone by spending money on hip arthroplasty candidates, rather than other conditions.
OHS, Oxford Hip Score; QALY, quality-adjusted life year.

Introducing this policy could cost the NHS an additional £36 million for each annual cohort of patients (of which £5.8 million would be due to additional surgical assessments), but would gain 30 000 QALYs. This policy would be highly cost-effective compared with current practice, costing just £1195 per QALY gained. Introducing OKS thresholds between 32 and 43 would produce still greater health benefits and cost less than £20 000 per QALY gained.

### Hips

Similarly, the ACHE study found that people with preoperative OHS ≤35 have a ≥70% chance of achieving an eight-point increase in OHS following hip arthroplasty.[4] Within the NOC hub, 87% (88 of 101) of patients had OHS ≤35, of whom 35 were referred. One patient was referred with OHS of 44. Introducing a threshold of 35 might therefore result in 26 additional referrals: a 30% increase.

Extrapolating across England, we might expect 14 000 additional surgical assessments and 5000 additional hip replacements each year if a threshold of 35 was introduced at the hub (table 2). The impact of this policy is lower than for knee osteoarthritis, as 90% of hip replacements are done in patients who were referred directly and did not attend the hub. The policy could cost an additional £25 million (of which additional surgical assessments account for £2 million) and gain 16 000 QALYs, costing £1560 per QALY gained. Introducing thresholds between 36 and 45 would produce greater health benefits and cost <£20 000 per QALY gained.

### DISCUSSION

This retrospective analysis characterised the patients being referred to and from a musculoskeletal hub, and showed that OKS/OHS and other factors were considered when deciding which patients to refer for surgical assessment. Since most patients attending hubs have relatively severe osteoarthritis symptoms and few patients with high OKS/OHS undergo arthroplasty (partly due to the low OKS/OHS thresholds used by some CCGs), introducing evidence-based OKS/OHS thresholds for arthroplasty is unlikely to prevent large numbers of inappropriate referrals. However, since recent evidence demonstrates that arthroplasty is both cost-effective and highly likely to benefit people with OKS/OHS well above the thresholds currently used by some CCGs,[4 5] such policies are also likely to identify patients who are not currently referred, but for whom arthroplasty could be expected to be beneficial and cost-effective. Extrapolating from the hub data suggests that setting thresholds of OKS ≤31 and OHS ≤35 would result in many more patients being referred for surgical assessment and surgery, although the exact patient numbers are uncertain and rely on assumptions. More evidence is therefore urgently needed on the other factors currently influencing both the decision to refer from the hub to surgical assessment and the decision on whether arthroplasty is the appropriate treatment option.

To our knowledge, our study is the first to report the characteristics of patients attending a musculoskeletal hub. However, the analysis was based on a small sample from only one hub and included only eight hub attendees with OKS/OHS ≥40. The proportion of patients undergoing surgery after surgical assessment is uncertain as

only 22 hub attendees underwent arthroplasty; however, varying the probability of undergoing surgery over its 95% CI did not materially change our conclusions. Data were analysed 1 month after the end of the 1-year period studied; some patients may therefore have gone on to have arthroplasty after physiotherapy or weight loss treatment after our data were extracted. Furthermore, the retrospective review of medical records may not have identified all patients meeting exclusion criteria, particularly for patients who did not attend either the hub or surgical assessment. Furthermore, no data were collected on BMI.

Estimates of the impact of different policies rely on additional assumptions and represent an approximate indication of potential patient numbers; prospective pilot studies would be required to assess the true impact in practice. In particular, we focused on the impact of changing referral guidelines at hubs, as no data were available on patients visiting GPs with osteoarthritis symptoms. In practice, any decision aid or referral guideline is also likely to affect GPs' referral decisions and the hub triage process, which could increase the number of additional operations resulting from any policy change. We assumed that decisions made by the patient and the surgeon about whether to proceed to surgery after surgical assessment would be unaffected by the use of thresholds by the hub as OKS/OHS would have already been taken into account at the hub; the budget impact for patients referred via the hub could be lower than shown in tables 1 and 2 if patients referred with high OKS/OHS were less likely to undergo surgery. We also assumed, arbitrarily, that 50% of people with OKS/OHS below the threshold who are not currently referred would choose not to be referred for surgery even if this were offered. The model also assumed that patients who do not undergo arthroplasty following their hub attendance would not have surgery for 10 years, since there are currently no data on how OKS/OHS change over time without arthroplasty.[5] In practice, many patients who would be eligible for surgery if the OKS/OHS threshold was raised would otherwise have had surgery later, after their condition had deteriorated. If the OKS/OHS thresholds were raised, the number of operations and budget impact might decrease over time as these patients would have been treated earlier, before their disease progresses, and would not need primary arthroplasty in the future.

Although access to joint replacement may be restricted if funding for interventions is reduced, our results suggest that increased numbers of knee/hip replacements could be justified on cost-effectiveness grounds. Our results suggest that the NHS is currently willing to pay no more £2000 per QALY gained from arthroplasty, while the National Institute for Health and Care Excellence routinely approves treatments for patients with other conditions (eg, rheumatoid arthritis) that cost £20 000[18] or even £40 000[19] per QALY. This suggests that it may be more efficient and more equitable to spend limited NHS funds on conducting more joint replacements, increasing

the operation rate in the UK to a level similar to that of Austria or Germany,[20] rather than directing these resources to other conditions. However, the number of procedures is also limited by availability of surgeons, operating theatres and hospital beds, which may mean that it is not currently feasible to conduct all of the operations that could be justified on cost-effectiveness grounds, and as noted earlier OKS/OHS is clearly not the only factor taken into account in decision-making by hubs and surgeons. In particular, patient choice plays a substantial role, and comorbidities and other factors are also taken into account.

Referral hubs have previously been shown to be an effective mechanism for referring patients to the most appropriate healthcare interventions/setting and patients are generally satisfied with this model of care.[8] One aim of introducing hubs was to identify patients who do not want surgery or who have mild symptoms and direct them to other effective interventions if surgery is not appropriate. However, the process and criteria for arthroplasty referrals still vary greatly between CCGs.[1] Explicit guidelines and decision aids could help reduce geographical inequity and ensure that all patients likely to benefit have fair access to cost-effective treatments. Although our analysis is based on retrospective data on a small sample from only one hub, it provides initial estimates of what the potential patient numbers and costs might be if a more explicit threshold-based approach was adopted nationally. While arthroplasty rates and patient characteristics in Oxfordshire are similar to the national average,[21 22] referral guidelines and pathways vary substantially between CCGs, suggesting that a larger study covering multiple hubs is necessary to provide a more comprehensive picture of current clinical practice. Ideally this would also collect more detailed patient information (eg, BMI or comorbidities).

In conclusion, our study demonstrates that musculoskeletal hubs currently take account of OKS/OHS and a variety of other factors when making decisions about referral to surgical assessment and arthroplasty. Introducing evidence-based thresholds for hip/knee replacement based on OKS/OHS, such as the ACHE tool, is likely to prevent very few inappropriate referrals, but identify many more patients who are not currently referred but for whom arthroplasty is likely to be beneficial and cost-effective. Our results can be used to estimate the impact of other policies, including those where thresholds vary by age or other patient characteristics. However, our estimates of potential patient numbers are approximations relying on numerous assumptions; a multicentre pilot study would be required to evaluate the actual impact of a policy change on routine clinical practice.

**Author affiliations**
[1]Health Economics Research Centre, Nuffield Department of Population Health, University of Oxford, Oxford, UK
[2]Max Planck Institute for Demographic Research, Rostock, Germany
[3]Nuffield Department of Orthopaedics, Rheumatology and Musculoskeletal Sciences, University of Oxford, Oxford, UK

⁴Centre for Healthcare Resilience and Implementation Science, Australian Institute of Health Innovation, Macquarie University, Sydney, New South Wales, Australia

**Acknowledgements**  We would like to thank the ACHE patient and public representatives, user group and steering group for their role in the project: Anne Clarkson Webb, Anthony Johnson, Chad Lion Cachet, Fiona Watt, Fraser Old, Gill Dean, Gillian Kempster, Jannie Kramer, Jennifer Bostock, Jiyang Li, Jo Hewanicka, John Nolan, Kate Jackson, Laura Ingle, Mary Snow, Matthew Cheetham, Patricia Mary Rubery, Sharon Barrington, Sujin Kang, Tim Wilton, Vida Field, Sunny Deo, Malvin Drakely, Jon Campion, George Peat, Peter Kay, Jon Waite and Sue Woollacott. We would like to extend thanks to Reza Mafi and Sarah Dorman, who helped with data cleaning. We would also like to thank the reviewers for their insightful comments on the manuscript and Emiko Sykes for proof-reading the manuscript. Data on NHS PROMs linked to HES APC data were reused with the permission of NHS Digital, copyright 2015, with all rights reserved. The following other members of the ACHE study group all contributed to the developing the research question, planning the protocol of work, gaining funding and/or delivery of this research project: Ray Fitzpatrick, David Murray, Nigel Arden, Andy Carr, Stephanie Smith, Kristina Harris, Rob Middleton, Elizabeth Gibbons, Elena Benedetto, Jill Dawson, Adrian Sayers, Laura Miller, Elsa Marques, Rachael Gooberman-Hill, Ashley Blom, Andrew Judge and Sion Glyn-Jones.

**Collaborators**  The ACHE study team: Sujin Kang, Jonathan Cook, Ray Fitzpatrick, David Murray, Nigel Arden, Andy Carr, Stephanie Smith, Kristina Harris, Rob Middleton, Elizabeth Gibbons, Elena Benedetto, Jill Dawson, Adrian Sayers, Laura Miller, Elsa Marques, Rachael Gooberman-Hill, Ashley Blom, Andrew Judge and Sion Glyn-Jones.

**Contributors**  AJP is the guarantor for this study. AJP, DB, HAD, AG and KLB conceived, designed and conducted the ACHE study. HAD, AJP and JS conceived and designed the study, which was conducted by HAD and JS, with assistance from Reza Mafi and Sarah Dorman. HAD, AG and PE conceived and designed the economic evaluation. HAD conducted the statistical and economic analyses under the supervision of AG. AJP, DB and KLB helped interpret the results. HAD drafted the manuscript. All authors edited the manuscript for important intellectual content and approved the final version.

**Funding**  The ACHE study and economic evaluation was funded by the NIHR Health Technology Assessment Programme (project number 11/63/01) and will be published in full in *Health Technology Assessment*. Further information is available at https://www.journalslibrary.nihr.ac.uk/programmes/hta/116301. This report presents independent research commissioned by the National Institute for Health Research (NIHR). AG was supported by the NIHR Biomedical Research Centre, Oxford. The views and opinions expressed in this publication are those of the authors and do not necessarily reflect those of the NHS, the NIHR, MRC, CCF, NETSCC, the Health Technology Assessment Programme, NHS, NHS Digital or the Department of Health. The funder had no involvement in the collection, analysis and interpretation of data; in the writing of the report; and in the decision to submit the article for publication. The authors had full access to the data (including statistical reports and tables), can take responsibility for the integrity of the data and the accuracy of the data analysis, and are responsible for submitting this paper.

**Competing interests**  HAD reports personal fees from Halyard Health, outside the submitted work. DB reports grants from Zimmer Biomet, outside the submitted work. AJP reports personal fees from Zimmer Biomet and DePuy, outside the submitted work. All declare that there are no other relationships or activities that could appear to have influenced the submitted work. The other authors declare that there are no relevant conflicts of interests or funding besides the NIHR grant that supports the current work.

**Patient consent for publication**  Not required.

**Ethics approval**  Trust management approval from the Oxford University Hospitals NHS Trust was obtained for retrospective analysis of anonymised data from medical records concerning outcomes in hip and knee arthroplasty (Oxford University Hospitals Research and Development Reference 11603). The plan to study referral patterns through the musculoskeletal hub, under the terms of the research and development agreement noted above, was discussed and agreed through the Clinical Governance Group for hip and knee replacement. The analysis was conducted to improve service understanding and used retrospective data on patients who had received normal clinical management. All data were anonymised by clinical staff.

**Provenance and peer review**  Not commissioned; externally peer reviewed.

**Data availability statement**  The decision tree and Markov models used in this paper are available from the corresponding author on request. PROMs/HES data are available from NHS Digital (http://content.digital.nhs.uk/dars).

**ORCID iDs**
Helen A Dakin http://orcid.org/0000-0003-3255-748X
Peter Eibich http://orcid.org/0000-0002-0689-0302
Alastair Gray http://orcid.org/0000-0003-0239-7278
Andrew J Price http://orcid.org/0000-0002-4258-5866

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
