## [Reviewer comments · BMJ Open]

ARTICLE DETAILS

TITLE (PROVISIONAL)	Who gets referred for knee or hip replacement? A theoretical model of the potential impact of evidence-based referral thresholds using data from a retrospective review of clinic records from an English musculoskeletal referral hub
AUTHORS	Dakin, Helen; Eibich, Peter; Gray, Alastair; Smith, James; Barker, Karen L.; Beard, David; Price, Andrew

VERSION 1 – REVIEW

REVIEWER	David Gwynne-Jones University of Otago, New Zealand
REVIEW RETURNED	22-Jan-2019

GENERAL COMMENTS	This paper has a lot of statistical modelling based on poor primary data. There is very limited data on the patient characteristics referred to the hip . Only age and gender has been collected. No mention of BMI, co-morbidities etc. OHS and OKS are only available on a very limited number of patients who attended the hub rather than on all who were referred. The problem facing public health services is excessive demand as briefly discussed in the intro. With the model described only around 50 of 315 patients referred got TKR. This suggests a lot of inappropriate referrals. If the first part of the paper was redone with more details preferably collected prospectively it may be of some interest. The second part is a sophisticated back of envelope calculation using the author own data to set a threshold that they believe is cost effective. This is more of a political message than a relevant clinical message. Some of it would be better placed in a discussion rather than results. I would be more interested to see the effect of modelling lower thresholds such as those mentioned in the intro as well as those used by the authors. I am not sure how generalisable these results are and to extrapolate to the whole NHS is a big step even in a theoretical paper. The idea that all patients should be referred based on an OHS or OKS of 31 or 35 seems ridiculous . From the authors own results there are only a few patients that have these higher scores with the majority undergoing surgery having much poorer scores . I agree with many of the sentiments expressed in the discussion about cost-effectiveness and recognise the special case that may require surgery despite a high score. I do not feel this paper adds much or is particularly relevant to clinical practice.
--

REVIEWER	Zoe Paskins Keele University, UK
REVIEW RETURNED	15-Mar-2019

GENERAL COMMENTS	This paper is generally well written and answers an important question. My concern is about the lack of ethical approval and description of this as an audit, and I feel I need more reassurances here, although would also welcome the handling Editor's view. The authors have sought advice on this and have R and D approval with a case reference. The abstract states clearly the design is an audit, with a decision tree model to evaluate policy change. The introduction describes a 'study' with aims to describe patient characteristics, identify management and estimate relationships between pre-op characteristics and referral decisions. While the second of these might relate to audit, the audit standard is not clear to me. The third aim feels like a research question, and as such I feel the lack of ethical approval may be an omission. In order to develop the decision tree 'consultations with staff' were also conducted. A STROBE checklist has also been completed. Depending on the editor's views, this may need a clearer focus on audit and the audit standard in both introduction, methods and results, and possibly evidence of the nature of the audit that R and D approved. The results section is also quite wordy and contains a few references 'anecdotally..' which have presumably arisen from the 'consultations with staff'. I have not appraised the results or appropriateness of the logistic regression analysis.
--

REVIEWER	Mark Pennington King's College London, UK
REVIEW RETURNED	18-Mar-2019

GENERAL COMMENTS	This paper addresses an important topic, the role of musculoskeletal hubs in the assessment of patients for hip and knee surgery and the impact of assessment criteria on the number of surgeries undertaken. I found the paper clear and well written, albeit the tables would benefit from better explanation. The methods are clearly described. In general, I thought the methods were appropriate, but the simplifications underlying the modelling of cost-effectiveness should have greater emphasis. The findings are an important counterweight to efforts to restrict access to orthopaedic surgery to save money. I thought the regression analysis of the impact of patient characteristics on the likelihood of referral from triage was appropriate. The sample for hips and knees was a little over 100 and the authors find the expected relationship between OHS/OKS and the likelihood of referral for surgical assessment. The authors also analyse the relationship between OHS/OKS and the likelihood of surgery once patients receive a surgical assessment. Here they find no relationship. This seems plausible given the initial screening these patients have received. Nonetheless, the sample size upon which this latter analysis is based is very small, only 27 hip patients. I wondered if this sample is simply too small to find any relationship?
--

	The regression analysis is used to estimate the impact of applying a national OHS/OKS threshold at musculoskeletal hubs. The authors estimate the cost impact and the QALY gains/losses arising from applying different thresholds. This analysis draws heavily on a previous paper which examines the relationship between OHS/OKS at surgery and the health gain from surgery, and a paper modelling the cost implications and QALY gains from surgery. Frustratingly, the latter paper is still in press. Hence it's difficult to judge exactly how the modelling of QALY gains in this paper has been undertaken. Some of the assumptions underpinning this analysis are quite strong. Assumption number 2 includes the assumption that OKS and OHS score do not impact the likelihood of surgery following surgical assessment. Whilst this is supported by the analysis undertaken by the authors, the previous caveats regarding the sample size for this analysis are relevant. It is a strong assumption that changing the number of hub triage patients who are referred for surgical assessment will have no impact on the likelihood each patient receives surgery after surgical assessment. The authors note this in the discussion but the implications of this assumption could be spelled out a little more clearly. More fundamentally, I think the cost-effectiveness modelling assumes that patients who are not referred for surgery do not subsequently re-present for assessment and possible surgery. My understanding is that osteoarthritis is a degenerative condition and that patients are likely to get worse over time. A higher OHS/OKS threshold may bring forward the date of operation for many patients rather than changing a once only decision on surgery. An earlier surgery data will generate health gains for patients over the additional time period after surgery. However, surgery at a younger age increases the risk of prosthesis failure requiring revision. So these health gains may be offset to some extent by an increased revision burden. Ideally, these factors would be modelled to provide a more robust estimate of the impact of changing hub referral thresholds on costs and QALY gains. I think the authors' much simpler approach is acceptable, but the limitations of this approach need to be more clearly spelt out. The authors conclude that a higher referral threshold would be lead to increased numbers of hip and knee replacements and that this increased investment would be cost-effective. Whilst I think the simplifications underpinning the analysis need to be clearly stated, the paper's findings are justified and convey an important message, namely that restricting access to orthopaedic surgery until patients have progressed to a severe state is not justified on cost-effectiveness grounds. Minor points Abstract – I wouldn't normally hyphenate quality adjusted Assumption – I wasn't clear what the source/justification for assumption 4 was. Where does the 50% come from? There's also a typo in the text for 4) '...others who are be unsuitable...' I think some caveats are needed around the statements in the discussion of the impact of higher OKS/OHS thresholds. The initial increase in numbers may be offset over time as some surgery that would have happened at a later date is brought forward. The same caveats apply to the inference that the NHS is prepared to pay no more than £2,000 per QALY for arthroplasty surgery. Will this conclusion still hold if the effect of lower thresholds is to delay rather than deny surgery? One sentence jarred with me. At the start of paragraph 5 in the discussion, 'Referral hubs are an effective mechanism for ensuring
--	--

	that patient receive cost-effective treatment...’ This point seems to be contradicted by the earlier inference that inappropriately low thresholds for referral are denying patients access to cost-effective surgery. Table 1 and 2. I really struggled with these tables. The key issue is that the tables relate only to patients attending the hub and not all hip and knee patients. It’s clear to me now but it took a while! I know it’s in the title but can the authors make this aspect of the data a little clearer. Table A4. I think it should be OKS/OHS in the variable column.
--	---

VERSION 1 – AUTHOR RESPONSE

Reviewer: 1

Reviewer Name: David Gwynne-Jones

Institution and Country: University of Otago, New Zealand

Please state any competing interests or state ‘None declared’: None declared

This paper has a lot of statistical modelling based on poor primary data. There is very limited data on the patient characteristics referred to the hip . Only age and gender has been collected. No mention of BMI, co-morbidities etc. OHS and OKS are only available on a very limited number of patients who attended the hub rather than on all who were referred.

Response: We recognise the limitations of the data (which relied upon information that was already collected in patients’ records) and have acknowledged these in the paper.

The problem facing public health services is excessive demand as briefly discussed in the intro.

With the model described only around 50 of 315 patients referred got TKR. This suggests a lot of inappropriate referrals.

The musculoskeletal hub is not dedicated to screening only those patients who are potentially suitable for joint replacement. The hub is a pathway management service designed to signpost the most appropriate treatment for a variety of patients and a variety of subsequent interventions. The outcomes from triage could be advice only, weight loss management and exercise, more formal physiotherapy, injection delivered in clinic, referral for a course of physiotherapy or (for a minority of the patients) referral for consideration of surgery.

Furthermore, although only 49 knee patients were referred for surgical assessment after a face-to-face visit at the hub, 71 patients were referred directly to surgical assessment (without a face-to-face attendance at the hub) and 78 had radiography, MRI, ultrasound, physiotherapy or injections (see Figure 1). A key part of the hub and the surgical assessment visit also involves giving patients the information they need to make an informed choice about whether or not surgery is appropriate. Some patients may, clinically, be appropriate candidates for arthroplasty, but may decide not to undergo surgery, or to delay surgery, following the hub visit or the surgical assessment visit.

If the first part of the paper was redone with more details preferably collected prospectively it may be of some interest.

Response: Unfortunately at this stage it is not feasible to undertake additional data collection, since the funding for the project has already ended, although this remains a priority for future research. In response to this comment, we have highlighted in the strengths and limitations section, and on pages 4 and 12 that a prospective pilot study is needed.

The second part is a sophisticated back of envelope calculation using the author own data to set a threshold that they believe is cost effective. This is more of a political message than a relevant clinical message. Some of it would be better placed in a discussion rather than results. I would be more interested to see the effect of modelling lower thresholds such as those mentioned in the intro as well as those used by the authors. I am not sure how generalisable these results are and to extrapolate to the whole NHS is a big step even in a theoretical paper.

The idea that all patients should be referred based on an OHS or OKS of 31 or 35 seems ridiculous . From the authors own results there are only a few patients that have these higher scores with the majority undergoing surgery having much poorer scores . I agree with many of the sentiments expressed in the discussion about cost-effectiveness and recognise the special case that may require surgery despite a high score. I do not feel this paper adds much or is particularly relevant to clinical practice.

Response: The current paper does not propose thresholds, but instead builds upon work recently conducted to estimate the first evidence-based OKS/OHS thresholds for arthroplasty (described in References 4 and 5). These thresholds are based on a detailed analysis of more than 400,000 patient records using quantile regression and decision-analytical modelling rather than back of an envelope calculations or political messages. Although these papers are not yet published, they have received positive reviews in a major orthopaedic journal and we hope that they will be published before or at the same time as this paper. We have asked the editors to send copies of these papers to all reviewers to aid them in the second round of review of this manuscript. References 4 and 5 demonstrate that patients with $OKS \leq 31$ or $OHS \leq 35$ have a $\geq 70\%$ chance of a meaningful benefit (a 7 or 8 point improvement in OKS or OHS, respectively), and that arthroplasty is highly cost-effective for patients with even higher scores. The current manuscript extends this previous work by estimating the budget impact of the estimated thresholds and incorporating the cost of referrals.

Even if these higher thresholds were adopted, not everyone with OKS/OHS below the threshold would be referred for arthroplasty, since some patients would be clinically unsuitable or have comorbidities that would make surgery risky or inappropriate, while others would choose not to undergo surgery following a detailed discussion with an orthopaedic surgeon about the risks and benefits. In our analysis, we assumed that 50% of patients with OKS/OHS below the threshold who are not currently referred, would be referred if new thresholds were introduced. We recognise that at present relatively few patients undergo surgery with scores of 31-35, although this is partly due to the lower thresholds that are used by clinical commissioning groups in the absence of evidence on what the appropriate threshold should be.

Tables 1 and 2 present results for the thresholds of 18 and 24 that are sometimes used by clinical commissioning groups, while the text focuses on the evidence-based thresholds estimated in reference 4. We feel that there is unlikely to be any value in presenting results for thresholds lower than 18, or making the results section longer by describing results for two different thresholds in the text.

We acknowledge that the extrapolation to the whole NHS is speculative, although since our previous work suggested that introducing higher thresholds would be beneficial to patients and cost-effective, it is necessary to have some estimates of the potential impact of this policy. We already acknowledge in the paper that a pilot study would be needed to definitively estimate the impact of any proposed policy change.

Reviewer: 2

Reviewer Name: Zoe Paskins

Institution and Country: Keele University, UK

Please state any competing interests or state 'None declared': None declared

This paper is generally well written and answers an important question.

My concern is about the lack of ethical approval and description of this as an audit, and I feel I need more reassurances here, although would also welcome the handling Editor's view. The authors have sought advice on this and have R and D approval with a case reference. The abstract states clearly the design is an audit, with a decision tree model to evaluate policy change. The introduction describes a 'study' with aims to describe patient characteristics, identify management and estimate relationships between pre-op characteristics and referral decisions. While the second of these might relate to audit, the audit standard is not clear to me. The third aim feels like a research question, and as such I feel the lack of ethical approval may be an omission. In order to develop the decision tree 'consultations with staff' were also conducted. A STROBE checklist has also been completed.

Depending on the editor's views, this may need a clearer focus on audit and the audit standard in both introduction, methods and results, and possibly evidence of the nature of the audit that R and D approved.

We apologise for any confusion about the ethical approval processes that preceded the work. The work described in this paper is a retrospective analysis of anonymised secondary data and involved no change to patients' care and no additional data collection beyond what had already been recorded in patients' medical records.

Our unit has local trust management approval to use anonymised routinely collected data to review the outcomes of our hip and knee patient pathway (11603). This data includes all data collected on the patient pathway that can be analysed anonymously, including pre-operative and post-operative Oxford Knee Scores. The plan to study referral patterns through the musculoskeletal hub, under the terms of the research and development agreement noted above, was discussed and agreed through the Clinical Governance Group for hip and knee replacement.

In response to this comment, we have also revised the ethical statements within the paper to give the information in the above paragraph. We have also sent evidence of the R&D approval and the discussion at the Clinical Governance Group to the editors in confidence.

Based on this comment we have avoided using the term "audit" to describe this work. We have also reworded the aims as "We aimed to review the characteristics of patients attending hub consultations, identify how patients within a hub are currently managed and explore how current referral decisions vary with preoperative characteristics".

The results section is also quite wordy and contains a few references 'anecdotally..' which have presumably arisen from the 'consultations with staff'.

Response: The narrative about how patients are triaged and assessed by the hub is based on data collected in the audit and the experience of the authors, one of whom runs the Oxford hub and conducts the vast majority of the triage, while another is an orthopaedic surgeon. In response to this comment, we have avoided use of the word "anecdotally" and have shortened the "current referral pathway" section where possible.

I have not appraised the results or appropriateness of the logistic regression analysis.

Reviewer: 3

Reviewer Name: Mark Pennington

Institution and Country: King's College London, UK

Please state any competing interests or state 'None declared': None declared

This paper addresses an important topic, the role of musculoskeletal hubs in the assessment of patients for hip and knee surgery and the impact of assessment criteria on the number of surgeries undertaken. I found the paper clear and well written, albeit the tables would benefit from better explanation. The methods are clearly described. In general, I thought the methods were appropriate, but the simplifications underlying the modelling of cost-effectiveness should have greater emphasis. The findings are an important counterweight to efforts to restrict access to orthopaedic surgery to save money.

Response: we thank the reviewer for his comments

I thought the regression analysis of the impact of patient characteristics on the likelihood of referral from triage was appropriate. The sample for hips and knees was a little over 100 and the authors find the expected relationship between OHS/OKS and the likelihood of referral for surgical assessment. The authors also analyse the relationship between OHS/OKS and the likelihood of surgery once patients receive a surgical assessment. Here they find no relationship. This seems plausible given the initial screening these patients have received. Nonetheless, the sample size upon which this latter analysis is based is very small, only 27 hip patients. I wondered if this sample is simply too small to find any relationship?

Response: We agree that the secondary analysis predicting whether or not patients undergo arthroplasty after surgical assessment is likely to be underpowered. In response to this comment, we have added a sentence on page 6 of the online supplementary material regarding the lack of power, although we have not made any changes to the main body of the text based on this change because that analysis was only presented in the appendix.

The regression analysis is used to estimate the impact of applying a national OHS/OKS threshold at musculoskeletal hubs. The authors estimate the cost impact and the QALY gains/losses arising from applying different thresholds. This analysis draws heavily on a previous paper which examines the relationship between OHS/OKS at surgery and the health gain from surgery, and a paper modelling the cost implications and QALY gains from surgery. Frustratingly, the latter paper is still in press. Hence it's difficult to judge exactly how the modelling of QALY gains in this paper has been undertaken.

Response: We have appended the unpublished papers (references 4 and 5) to our submission and have asked the editors to send them to reviewers.

Some of the assumptions underpinning this analysis are quite strong. Assumption number 2 includes the assumption that OKS and OHS score do not impact the likelihood of surgery following surgical assessment. Whilst this is supported by the analysis undertaken by the authors, the previous caveats regarding the sample size for this analysis are relevant. It is a strong assumption that changing the number of hub triage patients who are referred for surgical assessment will have no impact on the likelihood each patient receives surgery after surgical assessment. The authors note this in the discussion but the implications of this assumption could be spelled out a little more clearly.

Response: We assumed that the decisions made by the patient and surgeon about whether to proceed to surgery after surgical assessment would be unaffected by the use of thresholds by the hub as OKS/OHS have already been taken into account at the hub. However, we acknowledge that our preliminary analyses are not powered to rule out the possibility that patients with higher OKS/OHS might be less likely to be referred for surgery after surgical assessment even if the decision-making process at the surgical assessment was conducted in the same standardised and consistent way. If decisions made during surgical assessment consultations are influenced by OKS/OHS or the total number of referrals, a change in hub policy to refer patients with higher OKS/OHS may therefore

decrease the overall proportion of surgical assessment attendees who are referred for surgery. This would decrease the estimated budget impact, but is unlikely to have a large effect on the cost-effectiveness of any policy change.

In response to this comment, we have separated assumption 2 into two parts, to emphasise this assumption and rephrased the paragraph on pages 12-3 of the discussion to read:

“In particular, we focused on the impact of changing referral guidelines at hubs, as no data were available on patients visiting GPs with osteoarthritis symptoms. In practice, any decision aid or referral guideline is also likely to affect GPs’ referral decisions and the hub triage process, which could increase the number of additional operations resulting from any policy change. We assumed that decisions made by the patient and surgeon about whether to proceed to surgery after surgical assessment would be unaffected by the use of thresholds by the hub as OKS/OHS would have already been taken into account at the hub; the budget impact for patients referred via the hub could be lower than shown in Tables 1 and 2 if patients referred with high OKS/OHS were less likely to undergo surgery.”

More fundamentally, I think the cost-effectiveness modelling assumes that patients who are not referred for surgery do not subsequently re-present for assessment and possible surgery. My understanding is that osteoarthritis is a degenerative condition and that patients are likely to get worse over time. A higher OHS/OKS threshold may bring forward the date of operation for many patients rather than changing a once only decision on surgery. An earlier surgery data will generate health gains for patients over the additional time period after surgery. However, surgery at a younger age increases the risk of prosthesis failure requiring revision. So these health gains may be offset to some extent by an increased revision burden. Ideally, these factors would be modelled to provide a more robust estimate of the impact of changing hub referral thresholds on costs and QALY gains. I think the authors’ much simpler approach is acceptable, but the limitations of this approach need to be more clearly spelt out.

Response: We thank the reviewer for this insightful comment. Unfortunately, it is difficult to model the extent to which joint replacement may be delayed rather than avoided since there are no data on how OKS/OHS changes over time in the absence of arthroplasty. The best available data on WOMAC suggests that the proportion of patients whose WOMAC scores worsened was approximately equal to the proportion whose WOMAC scores improved (1-3). However, in our clinical experience, patients’ symptoms do tend to worsen over time and, as you suggest, commissioners setting low OKS/OHS thresholds may delay surgery rather than avoid it for many patients. Our previous work suggested that hip/knee replacement is cost-effective compared with no arthroplasty even for patients aged 50-60 years with OKS up to 43-44 an OHS up to 45: even when we allow for the cost and quality of life impact of revisions, and regardless of whether the time horizon was 10 or 60 years (reference 5).

In response to this comment, we have stated this assumption on page 8 and added 3 sentences on page 13 of the discussion to say: “The model also assumed that patients who do not undergo arthroplasty following their hub attendance would not have surgery for 10 years, since there are currently no data on how OKS/OHS change over time without arthroplasty.(4) In practice, many patients who would be eligible for surgery if the OKS/OHS threshold was raised would otherwise have had surgery later, after their condition had deteriorated. If the OKS/OHS thresholds were raised, the number of operations and budget impact might decrease over time as these patients would have been treated earlier, before their disease progresses, and would not need primary arthroplasty in the future.”

The authors conclude that a higher referral threshold would be lead to increased numbers of hip and knee replacements and that this increased investment would be cost-effective. Whilst I think the simplifications underpinning the analysis need to be clearly stated, the paper’s findings are justified

and convey an important message, namely that restricting access to orthopaedic surgery until patients have progressed to a severe state is not justified on cost-effectiveness grounds. We thank the reviewer for his comments. We hope that the assumptions are now clearly stated.

Minor points

Abstract – I wouldn't normally hyphenate quality adjusted

Response: This has been amended in the abstract and introduction

Assumption – I wasn't clear what the source/justification for assumption 4 was. Where does the 50% come from? There's also a typo in the text for 4) '...others who are be unsuitable...'

Response: The 50% figure is arbitrary and simply intended to give an indication of what the patient numbers might be. It was estimated by the clinical co-authors. In response to this comment, we have rephrased the first sentence of this paragraph to say "We assumed, based on the experience of clinical co-authors, that...". We also thank the reviewer for identifying the typo, which has now been corrected.

I think some caveats are needed around the statements in the discussion of the impact of higher OKS/OHS thresholds. The initial increase in numbers may be offset over time as some surgery that would have happened at a later date is brought forward.

Response: We thank the reviewer for raising this point. We hope that the sentences added on pages 12-3 address this point.

The same caveats apply to the inference that the NHS is prepared to pay no more than £2,000 per QALY for arthroplasty surgery. Will this conclusion still hold if the effect of lower thresholds is to delay rather than deny surgery?

Response: The answer to this question is far from clear and seems to depend on patient characteristics and assumptions about how quickly OKS/OHS might change over time (which, as discussed above, is something for which data are almost completely lacking). Based on our Markov model, for 70-year-old women with OHS of 24, doing hip arthroplasty immediately costs £2030/QALY (an incremental cost of £4428 ÷ a gain of 2.182 QALYs) compared with no arthroplasty over the next 10 years. If we imagine that the same women would have an OHS of 21 in three years' time, operating in 3 years' time would cost an additional £4050 and gain 2.014 QALYs over the 10 year time horizon compared with no surgery. Operating in three years' time once their OHS has fallen by three points would therefore cost £2011/QALY compared with no surgery. Operating now would cost £2250 compared with delaying by three years, suggesting that it is cost-effective to operate immediately rather than delay. For some other patient groups, delaying surgery is extendedly dominated. These sensitivity analyses are extremely speculative and rely upon crude assumptions about how quickly OKS/OHS might change and when we might consider undergoing surgery, so are not sufficiently robust to include in the paper. However, they do appear to suggest that the inference that the NHS is prepared to pay no more than £2,000 per QALY may hold regardless of whether surgery is delayed or denied.

One sentence jarred with me. At the start of paragraph 5 in the discussion, 'Referral hubs are an effective mechanism for ensuring that patient receive cost-effective treatment...' This point seems to be contradicted by the earlier inference that inappropriately low thresholds for referral are denying patients access to cost-effective surgery.

Response: We thank the reviewer for alerting us to this. In response to this comment, we have rephrased this sentence to read "Referral hubs are an effective mechanism for referring patients to the most appropriate healthcare interventions/setting and patients are generally satisfied with this model of care.(5)"

Table 1 and 2. I really struggled with these tables. The key issue is that the tables relate only to

patients attending the hub and not all hip and knee patients. It's clear to me now but it took a while! I know it's in the title but can the authors make this aspect of the data a little clearer.

Based on this comment, we have expanded the table legends by adding an additional sentence to make this point clearer – e.g. for Table 1: “The results presented exclude patients who did not attend face-to-face consultations at the hub; based on our analysis, 31% (24,063/76,617) of knee replacements are conducted on patients who attended the hub.”

Table A4. I think it should be OKS/OHS in the variable column.

Response: we thank the reviewer for identifying this error, which has now been corrected

VERSION 2 – REVIEW

REVIEWER	David Gwynne-Jones Dunedin School of Medicine, University of Otago, Dunedin, New Zealand
REVIEW RETURNED	14-Jul-2019

GENERAL COMMENTS	I continue to have problems trying to understand the message and relevance of this paper. Introducing threshold levels implies some form of rationing but this paper is suggesting use cost-effectiveness thresholds which will increase the demand. I found the paper difficult to follow and am not sure that the methods including assumptions in the model, results as presented and conclusions are fully justified. While it is a counter to the rationing argument I do not think the analysis is robust enough to satisfy policy makers. The first part of the paper describes the function of the hub and provides some limited descriptive details. There is no critique of whether the hub is functioning efficiently. I am surprised at the low conversion rate from surgical assessment to surgery in both the triage direct to surgical assessment group (56% knees, (69% hips is more acceptable)) and those referred from the hub (24% knees, 28% hips). This suggests that the triage function of the hub could be better. It appears that OHS/OKS was only collected on those seen face to face at the hub and that this score was one of the factors used to determine whether the patient was referred for surgical assessment. I assume that the introduction of a threshold score would only be used for those patients seen face to face to help determine referral. Why is OHS/OKS not used prior to hub triage to help determine who goes directly for surgical assessment etc. We have found it helpful for virtual triaging. Is assumption 2 page 7 valid? Is Assumption 6 valid? The authors present no data from their study but there is data available. 10 years is a long time to assume no change is a condition that is usually progressive. P10 line 15 130/315 = 41% not 44% The authors recognize the limitations of their data in their response but I think this is a major flaw. The short term follow up is also a major cause for concern. The authors state in line 44
--

	discussion that theirs is the first paper to describe the characteristics of patients attending a hub. With a better data set there would be scope for a paper on the functioning of the hub. Figure 1 and figure 2 need to be much clearer and include numbers, % and outcomes. Because of the weaknesses of part 1 there must be major concerns with the theoretical extrapolation to the whole of the NHS. There may be evidence to support the cost effectiveness of THR/TKR in patients with OHS/OKS <31. Providing TJR to appropriate patients with scores up to this level will obviously result in the need for more consultations and operations. This paper does not model a threshold of 31 for surgery as in the ACHE study but uses it as a threshold for referral for surgical consultation. This is not the same thing but does not seem to have been highlighted. The numbers of patients getting surgery in the hub study are too low to extrapolate the likelihood of surgery as pointed out by reviewer 3. I agree with a number of the other concerns raised by reviewer 3 and do not feel that they have been adequately addressed. Table 1 and 2 results could be commented on more clearly in the results section. Much of the results on page 11 refers to the ACHE study and should probably be in the discussion. It is unusual to cite references in a results section. Are the main results the authors want to convey the extra referrals, the cost-effectiveness, the need for extra surgery? Lots of results are presented in the results, tables and supplementary tables. Please consider highlighting the important ones. I think that possibly the paper should be split into two papers; one focusing on the hub results and a theoretical one using data from a number of sources to estimate the unmet demand for surgery if a surgical threshold of 31 was used.
--	--

REVIEWER	Mark Pennington King's College London
REVIEW RETURNED	12-Jul-2019

GENERAL COMMENTS	I am happy with the revisions to the manuscript. I noted one typo in the revisions - the second sentence under 'Hub collection data' the word was has been inappropriately deleted.
---

VERSION 2 – AUTHOR RESPONSE

Reviewer: 3

Reviewer Name: Mark Pennington

Institution and Country: King's College London

Please state any competing interests or state 'None declared': None declared

Please leave your comments for the authors below

I am happy with the revisions to the manuscript. I noted one typo in the revisions - the second sentence under 'Hub collection data' the word was has been inappropriately deleted.

Thank you for alerting us to this typo, which has now been corrected.

Reviewer: 1

Reviewer Name: David Gwynne-Jones

Institution and Country: Dunedin School of Medicine, University of Otago, Dunedin, New Zealand

Please state any competing interests or state 'None declared': none

I continue to have problems trying to understand the message and relevance of this paper.

The key messages of this paper are:

- Musculoskeletal hubs currently take account of Oxford knee and hip scores and a variety of other factors when making decisions about referral to surgical assessment and arthroplasty.
- Introducing evidence-based thresholds for hip/knee replacement based on OKS/OHS, such as the ACHE tool, is likely to prevent very few inappropriate referrals, but identify many more patients who are not currently referred but for whom arthroplasty is likely to be beneficial and cost-effective.

Furthermore, since no previous papers have described the patient characteristics of patients attending musculoskeletal hubs, our study is also valuable for understanding the clinical pathway at the hub and provides the only published evidence to inform cost-effectiveness models and budget impact calculations on interventions that may change which patients are referred for surgery. We are aware of the limitations of our dataset. Nevertheless, we think that these data are important and that the present paper makes the case for further data collection in this area.

In response to this comment, we have included the text from the above bullet points in the concluding paragraph.

Introducing threshold levels implies some form of rationing but this paper is suggesting use cost-effectiveness thresholds which will increase the demand.

Many clinical commissioning groups already "ration" access to hip and knee replacement based on patients' symptoms, either with or without use of patient-reported outcome measures, such as the Oxford hip/knee scores. As we discussed in the introduction, many UK clinical commissioning groups currently use very low thresholds, such as 19[1] or 24[1-3] out of 48, that are not based on robust evidence. We model what might happen if explicit evidence-based thresholds were used in place of any thresholds or assessments of symptoms that hubs are currently using. The ACHE study identifies an absolute threshold above which patients cannot achieve a 7/8-point improvement, a range of values from which decision-makers could select a "relative" threshold that would enable a particular proportion of patients to achieve a 7/8-point improvement, and economic thresholds above which arthroplasty is not cost-effective. In the current manuscript, we present budget impact estimates for a range of different thresholds, including those currently used by some CCGs and all three of the thresholds estimated in ACHE, but focus the text on results for a "relative" threshold that would ensure that 70% of patients achieve a 7/8-point improvement. Because all of the evidence-based thresholds estimated in the ACHE study are much higher than those that are used (implicitly or explicitly) at present, introducing these thresholds would prevent very few inappropriate referrals/operations, and is likely to increase the overall number of referrals for surgery.

In response to this comment, we have expanded the last sentence of the introduction to read "To illustrate how such data could be used to inform policy, we estimated the potential impact of a change

in referral criteria: namely basing referrals from the hub to surgical assessment on evidence-based OKS/OHS thresholds, rather than the hub's current referral criteria."

In several places (e.g. the intervention section of the abstract), we have also revised the wording to make it clear that we are evaluating the impact of introducing "evidence-based thresholds" (in place of the criteria hubs are currently using). We have also explicitly mentioned the existing use of low thresholds in the first paragraph of the discussion "Since most patients attending hubs have relatively severe osteoarthritis symptoms and few patients with high OKS/OHS undergo arthroplasty (partly due to the low OKS/OHS thresholds used by some CCGs), introducing evidence-based OKS/OHS thresholds for arthroplasty is unlikely to prevent large numbers of inappropriate referrals."

I found the paper difficult to follow and am not sure that the methods including assumptions in the model, results as presented and conclusions are fully justified.

In response to this comment, we have tried to make the methods, justification of assumptions, results and discussion of limitations clearer. A junior doctor has also reviewed the manuscript and made some suggestions on how we might make the paper clearer, which we have implemented: for example, adding subheadings in the "Effect of introducing referral thresholds" and using active rather than passive voice in several places in the manuscript.

While it is a counter to the rationing argument I do not think the analysis is robust enough to satisfy policy makers.

The authors recognize the limitations of their data in their response but I think this is a major flaw. The short term follow up is also a major cause for concern. The authors state in line 44 discussion that theirs is the first paper to describe the characteristics of patients attending a hub. With a better data set there would be scope for a paper on the functioning of the hub.

Because of the weaknesses of part 1 there must be major concerns with the theoretical extrapolation to the whole of the NHS. There may be evidence to support the cost effectiveness of THR/TKR in patients with OHS/OKS <31. Providing TJR to appropriate patients with scores up to this level will obviously result in the need for more consultations and operations.

This paper intends to present a descriptive analysis of a hub cohort, describe the way in which hub decisions are currently made and illustrate a potential use of the data by presenting an exploratory analysis estimating what theoretically might happen if evidence-based OKS/OHS thresholds were introduced.

The thresholds that we focus on in the paper (31 for OKS and 35 for OHS) were estimated in references 4 and 5 based on the best available UK data, which covered >400,000 operations. The methods and results of the analyses estimating these thresholds will be published in two separate papers that are currently under review in a leading orthopaedic journal.

We acknowledge in the discussion that our extrapolations to get the additional patient numbers are estimates that rely upon several assumptions and a very limited sample (both in terms of size, follow-up and geography); they should not be taken as robust policy advice, but more as an exploration of "potential" implications of introducing thresholds estimated previously.

We also agree with the reviewer that our dataset is not sufficient to assess whether the NOC musculoskeletal hub is functioning well, although this is not the objective of the current study (see next comment).

We agree that it would be useful to conduct a larger study on several hubs with longer follow-up, and to do a prospective pilot to test the exploratory findings of this study and assess the true impact of any proposed policy change. However, since no larger study has yet been done, our data currently represents the only available data on the characteristics of patients attending musculoskeletal hubs and the only data available to inform estimates of policy impact. As such, we feel that it is important to publish our results. Our study also highlights the importance of collecting these data and making it available for research.

In response to these comments, we have reiterated the limitations of the study in the last sentence of the discussion and explicitly stated the length of follow-up and use of only one centre in the second paragraph of the discussion in order that the limitations are clearly stated. We emphasised at the bottom of page 7 that the estimates of changes to the number of referrals are exploratory and removed the numbers of additional operations from the first paragraph of the discussion, and instead said “Extrapolating from the hub data suggests that setting thresholds of OKS ≤ 31 and OHS ≤ 35 would result in many more patients being referred for surgical assessment and surgery, although the exact patient numbers are uncertain and rely on assumptions.” We also used more tentative wording on page 12 (e.g. “might”/“could” to describe the potential impact of introducing thresholds).

The first part of the paper describes the function of the hub and provides some limited descriptive details.

There is no critique of whether the hub is functioning efficiently. I am surprised at the low conversion rate from surgical assessment to surgery in both the triage direct to surgical assessment group (56% knees, (69% hips is more acceptable)) and those referred from the hub (24% knees, 28% hips). This suggests that the triage function of the hub could be better.

Assessing whether the hub is functioning efficiently is not the aim of this study. Since that question was outside the scope of our study, we did not collect the additional data that would be necessary to make judgements about the hub’s performance, such as details of any other types of surgery performed, or patients’/surgeons’ views on the hub process and whether referral was appropriate.

As we discussed in the previous round of reviewers’ comments, the hub does not exclusively deal with potential arthroplasty candidates and signposts a variety of patients to the most appropriate treatment options, including advice, weight loss, exercise, physiotherapy, injections and surgery. Since there was no direct referral physiotherapy service in Oxfordshire at the time (which we have added on page 9), the hub received many referrals where the correct clinical pathway was physiotherapy. This is one of the reasons why significant numbers of patients were filtered out and did not convert to surgery. The 69% conversion rate for hip replacement that we observed in this study benchmarks with other services, although it is more difficult to compare conversion rates among hub referrals and knee patients, due to the smaller patient numbers. Furthermore, many of the patients attending surgical assessment consultations will have not been referred to the surgeon to consider arthroplasty, but to consider interventional radiology (such as hip block) or other types of surgery. Others may make an informed decision (based on discussions at the hub or surgical assessment) to not undergo surgery. Since the aim of the study was to estimate the numbers of patients undergoing arthroplasty, we did not routinely collect data on these other operations.

In response to this comment, we have stated on page 10 that “Those not undergoing arthroplasty may have interventional radiology, or other operations” and on page 9 that the hub “also directed patients to physiotherapy in the absence of a direct referral physiotherapy service”.

We have also deleted the sentence “If the service works well, only patients who are suitable for surgery are referred” from page 14 of the discussion, to avoid giving a false impression of conversion rates and the diverse reasons for surgical assessment consultations. The paragraph now reads: “Referral hubs have previously been shown to be an effective mechanism for referring patients to the most appropriate healthcare interventions/setting and patients are generally satisfied with this model of care.[4] One aim of introducing hubs was to identify patients who do not want surgery or who have mild symptoms and direct them to other effective interventions if surgery is not appropriate.”

It appears that OHS/OKS was only collected on those seen face to face at the hub and that this score was one of the factors used to determine whether the patient was referred for surgical assessment. I assume that the introduction of a threshold score would only be used for those patients seen face to face to help determine referral.

In principle, any new threshold could be used by GPs in primary care, during the hub triage, during face-to-face consultations at the hub and/or in consultations with orthopaedic surgeons. However, we were unfortunately only able to collect data on OHS/OKS completed during face-to-face hub consultations. We therefore focused on estimating the budget impact for the subset of patients who attend face-to-face hub consultations and acknowledge in the third paragraph of the discussion that

thresholds may be used at other stages of the patient pathway and that their introduction may therefore have a greater impact on numbers of operations and costs than is estimated in this study.

In response to this comment, we have added a statement on pages 7-8 to say “Although in practice thresholds could be used at various stages in the referral pathway, our analysis focused on the impact of changing referral criteria during face-to-face hub consultations since OKS/OHS data were only routinely available for this patient group.” We also reiterated this in the first sentence of the “Effect of introducing referral thresholds” section: “We used the hub data to estimate the impact that using OKS to guide referral decisions during face-to-face hub consultations might have on patient numbers, costs and health outcomes.”

Why is OHS/OKS not used prior to hub triage to help determine who goes directly for surgical assessment etc. We have found it helpful for virtual triaging. As stated in the second paragraph of the Results section, OHS/OKS questionnaires are frequently completed by patients before they are referred to the hub and this information is currently used as part of the hub triage to determine which patients are referred directly for surgical assessment and which attend face-to-face consultations at the hub. Unfortunately, we were not able to quantify the impact of using evidence-based OHS/OKS thresholds during the hub triage since we did not have data on OHS/OKS for all patients referred to the hub. We hope that the statement added in response to the previous comment makes this clear.

Is assumption 2 page 7 valid?

In practice, this assumption is unlikely to hold, since the staff undertaking the hub triage explicitly consider OKS/OHS and (to a lesser extent) age and sex when deciding whether patients should attend a face-to-face consultation at the hub, be managed in primary care, or be referred directly for surgical assessment. We acknowledge this limitation on page 8. However, since there are no data on OHS/OKS for patients who did not attend face-to-face consultation at the hub, there is no way to empirically test this assumption or any evidence on which to make a more realistic assumption. This is one of the reasons why our analysis focuses on the sample of patients who attend face-to-face consultations at the hub: this simplifying assumption has negligible impact on our estimates of the budget impact for those patients who do attend the hub.

In response to this comment, we have acknowledged on page 8 that this assumption may not hold in practice: “Although this assumption may not hold in practice (as symptom severity is one of the main factors considered in the hub triage), it is unlikely to affect estimates of the impact of changing referral criteria solely at the hub”.

Is Assumption 6 valid? The authors present no data from their study but there is data available. 10 years is a long time to assume no change is a condition that is usually progressive.

We agree that this is a strong assumption. However, we are aware of no published data on the rate at which patients are referred back to clinic or on how OKS/OHS changes over time in the absence of arthroplasty. The best available data on WOMAC suggests that the proportion of patients whose WOMAC scores worsened was approximately equal to the proportion whose WOMAC scores improved [5-7]. However, in our clinical experience, patients’ symptoms do tend to worsen over time and, as you suggest, commissioners setting low OKS/OHS thresholds may delay surgery rather than avoid it for many patients. Nonetheless, in the absence of quantitative data, it is difficult or impossible to model the extent to which joint replacement may be delayed rather than avoided. However, in our previous economic evaluation varying the time horizon between 5 and 10 years had negligible impact on the cost-effectiveness of hip/knee replacement (see reference 5).

In the previous round of revisions, we therefore made sure that this assumption was explicitly stated on page 8 and discussed the implications in the discussion: for example, we say on page 14 that “In practice, many patients who would be eligible for surgery if the OKS/OHS threshold was raised would otherwise have had surgery later, after their condition had deteriorated. If the OKS/OHS thresholds were raised, the number of operations and budget impact might decrease over time as these patients would have been treated earlier, before their disease progresses, and would not need primary arthroplasty in the future.”

P10 line 15 $130/315 = 41\%$ not 44%

Thank you for alerting us to this typo, which has now been corrected.

Figure 1 and figure 2 need to be much clearer and include numbers, % and outcomes.

Figures 1 and 2 are intended to show the number of patients included in the analysis at each stage in accordance with the STROBE guidelines for observational studies. Percentages of patients and OKS/OHS are already shown in Figure 3 and we feel that including this information in Figures 1 and 2 would deviate from the style typically used for STROBE or CONSORT diagrams, make these figures harder to read and replicate information that is already given in Figure 3.

However, we have attempted to make these figures clearer by adhering more closely to the style used in the elaboration statements for CONSORT and STROBE, which we hope will make them easier to understand. We have also added a reference to Figure 3 in the first paragraph of the Results section to direct readers' attention to the additional information shown in figure.

We would welcome an opinion from the editor regarding whether Figures 1 and 2 are sufficiently clear, or whether they should be changed.

This paper does not model a threshold of 31 for surgery as in the ACHE study but uses it as a threshold for referral for surgical consultation. This is not the same thing but does not seem to have been highlighted.

The distinction between thresholds for referral and thresholds for surgery is indeed important. The ACHE study gives values for the OKS and OHS showing the likelihood of patients benefiting from arthroplasty. As correctly stated, the ACHE study estimates that patients with pre-operative OKS ≤ 31 have a $\geq 70\%$ chance of achieving a seven-point increase in OKS following knee arthroplasty. The ACHE thresholds are intended to support the shared decision-making process for referral to secondary care and may also help, but not replace, the complex shared decision-making process during surgical assessment consultations. The potential benefit of undergoing surgery and the appropriateness of being referred for consideration are therefore implicitly linked. We therefore don't feel that it is necessary to highlight this distinction in the paper.

In response to this comment we have clarified in the first paragraph of the introduction that the ACHE study estimated referral thresholds.

The numbers of patients getting surgery in the hub study are too low to extrapolate the likelihood of surgery as pointed out by reviewer 3. I agree with a number of the other concerns raised by reviewer 3 and do not feel that they have been adequately addressed.

We agree that the logistic regression analysis predicting the odds of patients undergoing surgery conditional on OKS/OHS, age and sex is underpowered and cannot rule out the possibility that these variables affect the odds of patients who have been referred to surgical assessment subsequently undergoing surgery. This is one of the reasons why we put this secondary analysis in the appendix and caveat the presentation of results with an acknowledgement that it is underpowered (Online supplementary file, page 5). The results of this logistic regression analysis are not extrapolated to predict national data.

Our extrapolations do base the overall likelihood of surgery on the 30% (12/40) of knee patients and 37% (10/27) of hip patients who underwent or were awaiting arthroplasty surgery after being referred to surgical assessment from the hub. We acknowledge that these patient numbers are modest. For hip replacement, the 95% confidence interval around the proportion of patients undergoing surgery in our hub dataset is 19% to 55%. If the probability of surgery had been 19%, the current number of operations predicted to be done on hub attendees would have been 3554 and introducing a fixed threshold of 35 would have increased the number of operations to 6284. By contrast, if the probability of surgery had been 55%, the current number of operations on hub attendees would have been 9333, which would have increased to 16,605 with a fixed threshold of 35. However, the percentage increase in the number of operations resulting from a policy change would have been virtually identical in all cases. Furthermore, the uncertainty around the proportion of patients undergoing surgery has no impact on the conclusion that introducing an evidence-based OHS threshold would increase the number of referrals and hip replacements and would be cost-effective.

We would welcome the editor's view on whether the comments from reviewer 3 have been adequately addressed, since Reviewer 3 commented that he was happy with the revisions to the manuscript.

In response to this comment, we have acknowledged the small numbers of patients undergoing surgery in the second paragraph of the discussion: "The proportion of patients undergoing surgery after surgical assessment is uncertain as only 22 hub attendees underwent arthroplasty; however, varying the probability of undergoing surgery over its 95% confidence interval did not materially change our conclusions." In the penultimate paragraph of the discussion, we also acknowledge the small sample and say that a larger study is needed.

Table 1 and 2 results could be commented on more clearly in the results section.

In response to this comment, a junior doctor has reviewed the manuscript and made suggestions on how we might make this section clearer, which we have implemented. In particular, we have explained the choice of thresholds and justified focusing on a knee threshold of 31 in the first paragraph of the "Effect of introducing referral thresholds" section. We have also added subheadings in the "Effect of introducing referral thresholds" section, used active rather than passive voice in several places, removed the references to the "economic threshold" and added footnotes in the tables to explain the net health benefit calculation, rather than presenting net health benefit in the text.

Much of the results on page 11 refers to the ACHE study and should probably be in the discussion. It is unusual to cite references in a results section.

Two sentences of the results section on page 11 of the original manuscript referred to the ACHE study (reference 4). A further two sentences referred to the ACHE cost-effectiveness paper (reference 5), reiterating the methods and stating that the economic threshold calculated in the ACHE study was 43. The remainder of the results section describes the results of the current study.

We acknowledge that citing references in the results section is unusual, although in this instance, we feel that citing reference 4 is essential in order to justify focusing on these thresholds rather than others.

However, we have removed both references to the ACHE cost-effectiveness paper (reference 5). We have also rephrased the second sentence of the "Effect of introducing referral thresholds" section to read "Table 1 shows results for thresholds between 18 and 43, although we focus here on the impact of an OKS threshold of 31, since the ACHE study reported that people with scores ≤ 31 have a $\geq 70\%$ chance of achieving a seven-point increase in OKS following knee arthroplasty.[4]" We hope that these changes make it clear which results are from reference 4 and which of the outcomes of the current study.

Are the main results the authors want to convey the extra referrals, the costeffectiveness, the need for extra surgery? Lots of results are presented in the results, tables and supplementary tables. Please consider highlighting the important ones.

The main results that we want to convey are the descriptive information about the characteristics of patients and decision-making at the hub, as well as the estimates of the number of additional referrals and the number of additional operations. However, we feel it is also informative to present the costs and health benefits alongside the numbers of operations to give a complete picture. We also feel that it is necessary to talk through each of the different types of results that are presented in Tables 1 and 2, rather than only discussing the numbers of referrals and operations in the text and leaving the reader to interpret the data on costs and QALYs without any additional explanation.

Rather than focusing on specific rows of Tables 1 and 2, we therefore focused on specific columns and talk through each of the results within Tables 1 and 2 for the thresholds that were estimated in the ACHE study. We hope that the changes we have made in response to the previous comment help make this clearer.

In response to this comment, we have also removed the net health benefits from the text (leaving them in the tables with additional explanation in footnotes) and clarified in the first paragraph of the "Effect of introducing referral thresholds" section that the thresholds are only applied to patients attending face-to-face consultations at the hub.

I think that possibly the paper should be split into two papers ; one focusing on the hub results and a theoretical one using data from a number of sources to estimate the unmet demand for surgery if a surgical threshold of 31 was used.

We feel that methods and results are most usefully presented in a single paper. Since we are not aware of any other UK datasets that would allow us to estimate how patient numbers might change with different referral thresholds, we feel that it is better to present the results from the hub together with the extrapolation. In particular, the extrapolation to the NHS shows the potential for analyses using the hub data and makes a case for collecting data from a wider range of hubs and making these data accessible for research. Similarly, the hub results provide important context for the extrapolation exercise and the potential limitations of this analysis. Therefore, we would strongly prefer to present both analyses as part of the same paper. However, if the editors would prefer to split the present paper in two, then we would of course be willing to make the necessary changes. We have nonetheless attempted to make the paper clearer and simplify it where possible (e.g. omitting net health benefit from page 12).

References

1. The Royal College of Surgeons of England. Is access to surgery a postcode lottery? 2014. <https://www.rcseng.ac.uk/news-and-events/media-centre/press-releases/many-ccgs-are-ignoring-clinical-evidence-in-their-surgical-commissioning-policies/> (accessed 8 November 2016).
2. Harrogate and Rural District Clinical Commissioning Group. Clinical thresholds: Hip and knee arthroplasty for osteoarthritis (only). 2014. <http://www.harrogateandruraldistrictccg.nhs.uk/data/uploads/rss2/hip-and-knee-arthroplasty.pdf> (accessed 8 November 2016).
3. Scarborough and Ryedale Clinical Commissioning Group. Hip replacement pathway. 2015. <http://www.scarboroughryedaleccg.nhs.uk/data/uploads/rss2/orthopaedics/hip-replacement-march-2015.pdf> (accessed 8 November 2016).
4. Candy E, Haworth-Booth S, Knight-Davis M. Review of the Effectiveness of a Consultant Physiotherapy-Led Musculoskeletal Interface Team: A Welsh Experience. *Musculoskeletal Care* 2016;14(3):185-91 doi: 10.1002/msc.1122[published Online First: Epub Date].
5. Holla JF, van der Leeden M, Heymans MW, et al. Three trajectories of activity limitations in early symptomatic knee osteoarthritis: a 5-year follow-up study. *Ann Rheum Dis* 2014;73(7):1369-75 doi: 10.1136/annrheumdis-2012-202984[published Online First: Epub Date].
6. Sharma L, Cahue S, Song J, Hayes K, Pai YC, Dunlop D. Physical functioning over three years in knee osteoarthritis: role of psychosocial, local mechanical, and neuromuscular factors. *Arthritis Rheum* 2003;48(12):3359-70 doi: 10.1002/art.11420[published Online First: Epub Date].
7. van Dijk GM, Veenhof C, Spreeuwenberg P, et al. Prognosis of limitations in activities in osteoarthritis of the hip or knee: a 3-year cohort study. *Arch Phys Med Rehabil* 2010;91(1):58-66 doi: 10.1016/j.apmr.2009.08.147[published Online First: Epub Date].